# Depression and its associated factors among health care workers in Saint Paul's hospital millennium medical college, Ethiopia

**Melese Bahiru Tesema**[1☉]*, **Berhanu Teshome Woldeamanuel**[2☉], **Eyoel Berhane Mekonen**[3‡], **Kidest Getu Melese**[4‡]

**1** Department of Primary Health Care Unit, Abebe Bikila Health Center, Addis Ketema Sub-City, Addis Ababa, Ethiopia, **2** Department of Epidemiology and Biostatistics, School of Public Health, Saint Paul's Hospital Millennium Medical College, Addis Ababa, Ethiopia, **3** Public Health Department, Saint Paul's Hospital Millennium Medical College, Addis Ababa, Ethiopia, **4** School of Midwifery, Wolaita Sodo University, Wolaita, Ethiopia

☉ These authors contributed equally to this work.
‡ EBM and KGM also contributed equally to this work.
* melesebahiru21@gmail.com

**Data Availability Statement:** All relevant data are within the manuscript and its Supporting Information files.

## Abstract

### Background

Depression is a common mental disorder that affects 3.8% of the general population and 24% of healthcare workers globally. Healthcare professionals are more susceptible to depression because they face higher amounts of professional stress in their jobs and academic lives. However, there is limited knowledge regarding health professionals' level of depression in Ethiopia. This study aimed to assess the prevalence of depression and associated factors among health professionals, at Saint Paul's Hospital Millennium Medical College, Addis Ababa, Ethiopia.

### Methods

We conducted an institution-based cross-sectional study among 439 randomly selected healthcare workers using interviewer-administered patient health questionnaire-9 from April to May 2023. Ordinal logistic regression was performed to identify predictors of depression. Reported $p$-values < 0.05 or a 95% Confidence Interval of Odds Ratio excluding one was considered statistically significant.

### Result

The overall prevalence of depression among healthcare workers was 21.9% (95%CI: 18%, 27.76%). There were reports of mild (35%), moderate (13%) and severe (9%) depression, respectively. Marital status of being single (AOR = 7.78, 95%CI: 1.123, 49.01), history of childhood abuse (AOR = 2.57, 95%CI:1.49, 4.42), history of suicidal attempt (AOR = 2.66, 95%CI:1.25,5.67), having a history of stressful life event (AOR = 1.527, 95%CI: 1.02,2.3), back pain over the past 30 days (AOR = 2, 95%CI: 1.30,3.11), working for more than 8 hours (AOR = 3.03, 95%CI: 1.12,8.24), and having experience of 5–10 year (AOR = 4, 95%

**Funding:** The author(s) received no specific funding for this work.

**Competing interests:** The authors have declared that no competing interests exist.

**Abbreviations:** COVID-19, Corona Virus Disease 2019; HCWs, Health Care Workers; LRT, Likelihood Ratio Test; PHQ-9, Patient Health Questionnaire-9; SPHMMC, Saint Paul's Hospital Millennium Medical College; VIF, Variance Inflation Factor; YLD, Years Lived with Disability.

CI: 1.05,15.27) and 10–15 years (AOR = 4.24, 95%CI: 1.08,16.58) and poor social support (AOR = 2.09, 95%CI: 1.09,3.99) were statistically associated with increased level of depression.

## Conclusion

Healthcare professionals' higher rate of depression was due to the higher workload, childhood abuse, history of stressful life, back pain, and poor social support. Thus, the hospital should give special attention to early screening and treatment for depression for those healthcare workers who have a high workload, childhood abuse, back pain, a history of stressful life and poor social support. Similarly, the Ministry of Health should also design strategies to screen, detect and treat depression among healthcare workers.

## Background

Depression is characterized by sadness, lack of interest or enjoyment, feelings of guilt or low self-worth, disturbed sleep or appetite, severe weight loss, fatigue, poor focus, psychomotor agitation or retardation, and recurring thoughts of death (suicide) [1]. Key signs of depression include low mood and a lack of interest [2]. Depression has consistently been among the top ten leading causes of all years lived with disability (YLDs) worldwide and the second leading cause of global YLDs, accounting for 5.6% of all YLDs in 2019 [3]. Individuals with a diagnosis of depression had a 40–60% higher probability than the general population to die at a younger age [4].

Studies have shown that healthcare workers (HCWs) are more likely than the general population to be exposed to depression due to the demanding nature of their occupations [5–7]. The prevalence of depression among HCWs varies from 21.53% to 32.77% in developed nations [8–12]. A meta-analysis and systematic review utilizing 57 cross-sectional studies conducted between 2019 and 2020 found that the pooled prevalence rate of depression among HCWs was 24% [13]. A comparable study conducted in 2020 in Wuhan, China, using the Patient Health Questionnaire-9 (PHQ-9) instrument among 994 medical professionals working in a high-risk environment, revealed that 36% of medical staff had sub-threshold depression, 34.4% had mild depression, 22.45% had moderate depression, and 6.2% had severe depression [14].

A study done in a teaching hospital in Nigeria shows the prevalence of depression to be 17.3% among resident doctors and 1.3% among non-resident doctors [5]. A similar study done in a teaching hospital in Egypt using PHQ-9 among physicians and nurses revealed a 71.4% prevalence of depression [6]. Further, in Kenya, 53.6% of HCWs had a positive test result for depression, with most of them having moderate depression [7]. Furthermore, studies done using Depression anxiety stress scale-21 among HCWs in Botswana, Uganda, and Sudan revealed prevalence rates of depression to be 21%, 40.3%, and 75%, respectively [15–17].

In Ethiopia, a systematic review and meta-analysis study reported a pooled prevalence rate of depression among HCWs was 11% [18]. In eastern Ethiopia, the prevalence of minimal, mild, moderate, and severe depression among HCWs was 27.9%, 24.1%, 9.4%, and 1.1%, respectively [19]. Another study in central Ethiopia [20], southern Ethiopia [21], and Gondar [22] reported the highest prevalence of depression among HCWs, 60.3%, 50.1%, and 42.7% respectively. The common risk factors associated with depression among HCWs are the clinical specialty, presence of chronic illness, substance use, history of psychiatric disorder,

insufficient social support, family history of mental illness, being unmarried, low income, long working hours, being female and working at night time [3].

Most of the studies in Ethiopia are focused on depression during the era of COVID-19 among those who are frontline HCWs for COVID-19 [20–22] and there is limited study done among HCWs in general. Healthcare workers working in general set-up (non-front line) are susceptible to mental illness specifically depression. However, research done among healthcare workers on depression among those whose working conditions as non-frontline (general healthcare setup) is limited. Therefore, this study aims to assess the prevalence of depression and its associated factors among healthcare workers in Saint Paul's Hospital Millennium Medical College (SPHMMC), Addis Ababa, Ethiopia.

## Methods

### Study design and setting

Institution based cross-sectional study was conducted from April to May 2023 at SPHMMC, Addis Ababa, Ethiopia. SPHMMC, as it is known today, was established through a decree of the council of ministers in 2010, although the medical school was opened in 2007 and the hospital was established in 1968 by the late Emperor Haileselassie I [23]. It is governed by a board under the Federal Ministry of Health. The hospital provides preventive, curative and rehabilitative care for patients coming from the health center by referral system. The inpatient capacity of the hospital is more than 700 beds, and the college sees an average of 1200 emergency and outpatient clients daily [23].

### Sample size and study population

Sample size calculations are used to determine either the sample size required to give a specified power or the power implied by a specified sample size. Thus, sample size calculations need to be undertaken prior to a study to avoid both the wasteful consequences of under-powering or overpowering. Further, in this study, we planned to perform a test of hypothesis comparing the proportions, with an ordered categorical outcome, where analysis is by the proportional odds model. Studies that use small samples not being able to demonstrate differences or associations that do exist. Such studies are 'under-powered', not possessing sufficient statistical power to detect the effects they set out to detect. The statistical power of a study builds in a safety margin to avoid generalizing false positive results which could have expensive consequences. A non-significant finding of a study may thus simply reflect the inadequate power of the study to detect differences or associations at levels that are conventionally accepted as statistically significant [24]. In this study, the target power is 80% or 0.8, meaning that a study has an 80% likelihood of detecting an association that exists. Furthermore, we used a double population proportion formula to calculate sample size with 95% CI, by using state calc of Epi-info. Thus, taking the ratio of unexposed to exposed, odd ratio and proportion of respective exposure variable and adding a 10% non-response rate the optimum sample size was 439 (Table 1).

The study participants were selected through a stratified random sampling method by taking profession as their strata. The sample size was allocated proportional to the size of a strata and finally, samples were selected using a simple random sampling technique within a strata (Fig 1). (Fig 1).

### Measurement of study variables

**Dependent variable.** Depression was the outcome variable of this study. The PHQ-9 was used to assess depression with nine items of major depressive disorder symptoms from

**Table 1. Assumption used to calculate sample size by double population proportion using stat-calc of Epi-info.**

| S.no | Variable as exposures | Assumptions | | | | | Sample size | Final sample size by adding 10% non-response rate |
|---|---|---|---|---|---|---|---|---|
| | | OR | P | Ratio | Power | CI | | |
| 1 | Sex (Female) | 1.94 | 24.5% | 1.83 | 80% | 95% | 395 | 435 |
| 2 | Marital status (single) | 2.16 | 22.1% | 0.70 | 80% | 95% | 296 | 326 |
| 3 | Family history of mental illness | 7.31 | 24.4% | 13 | 80% | 95% | 146 | 161 |
| 4 | Smoker | 2.67 | 22.2% | 3 | 80% | 95% | 222 | 244 |
| 5 | Workload (Hours) | 1.997 | 69 | 1.16 | 80% | 95% | 399 | 439 |

DSM-IV criteria. Each item response was rated as 0 = "not at all" to 3 = "nearly every day". The total score of nine items of PHQ-9 ranges from 0 to 27. Scores were defined as 0–4 no depression, 5–9 mild, 10–14 moderate, and 15 or above severe depression, respectively (26). PHQ-9 was validated previously with a sensitivity of 86% and specificity of 67% in the Ethiopian population [25]. For this study, the cut point of 10 or above was used as depression and below 10 no depression.

**Independent variables.** The exposure variables were sociodemographic variables (age, sex, marital status, religion, level of education, and monthly income), and psycho-social factors (history of childhood abuse, stressful life events, history of suicidal attempt, presence of family history of mental illness). Further, physical health status (history of injury, presence of disability, headache over past 30 days, back pain over the past 30 days, fever over past 30 days, medical problem, smoking history, and social support), health professional status, and job satisfaction level (field of study, working hour, work experience, work environment, working department, and job satisfaction) were used.

**Data collection method and tool.** The data was collected using interviewer administrated Questionnaire. Two trained health professionals were recruited to collect data and two of the authors supervised the data collection. Before the data collection pre-test with 5% of health professionals was conducted at Ras Desta Damtew Referral Hospital located near SPHMMC.

Sociodemographic variables, psycho-social factors, physical health status, and health professional status were assessed by HCW's response to yes/no options and among the listed

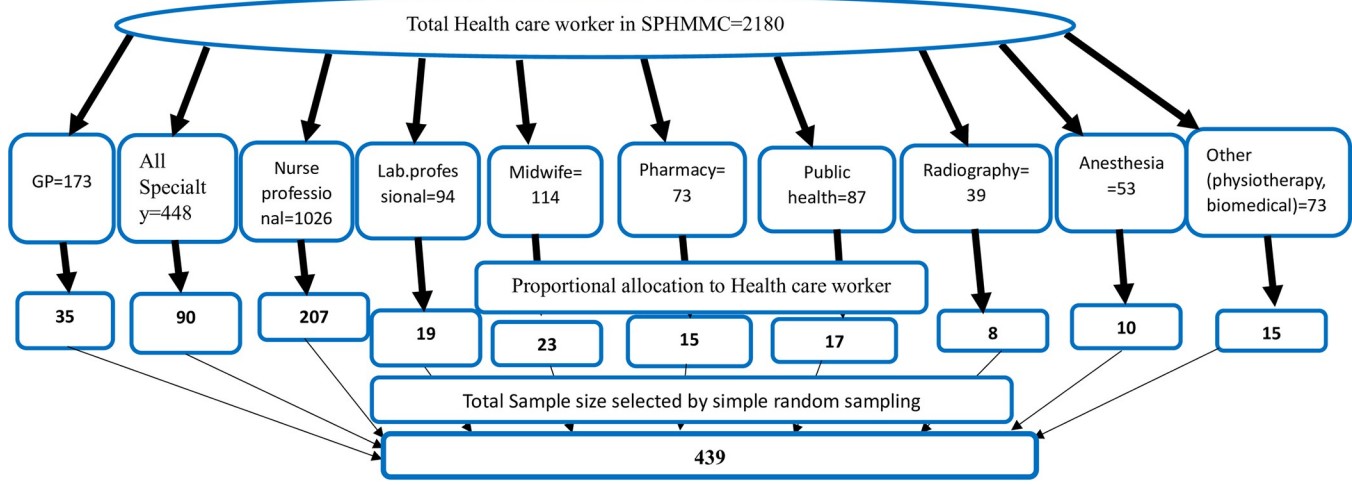

**Fig 1. Proportional allocation to study the prevalence of depression and associated factors among health care professionals working at SPHMMC, Addis Ababa, Ethiopia.**

categories for each question. To assess social support Oslo-3-item social support scale was used and those with scores of 3–8 were classified as poor social support, 9–11 as moderate social support, and 12–14 as strong social support [26]. Job satisfaction was measured by using short form twenty items Minnesota Satisfaction Questionnaire each scored a 5-point Likert scale with 1 denoting strongly dissatisfied and 5 denoting strongly satisfied. Overall job satisfaction was estimated by taking the sum score of all the subscales. HCWs who scored a value of 62 and below were considered as dissatisfied and those who scored greater than 62 were considered as satisfied [27]. Completeness and consistency of items were ensured to minimize systematic error.

## Statistical analysis

The data was checked and cleaned in Microsoft Excel version 19 and then exported to Statistical Package for Social Science version 26 for coding and analysis. A descriptive summary was used to present the study results. The categories of the response variable (depression level) are four, which are ordered as "no depression", "mild depression", "moderate depression" and "severe depression". Hence ordinal logistic regression has potentially greater power than that of binary logistic regression and the baseline-category logit models as it takes into account information on the order of values. The proportional odds model was used as a tool to model HCW's depression status because the outcome variable has a natural ordering by defining the cumulative probabilities differently rather than considering the probability of a single event. The main assumption under the proportional odds model was that it is invariant the regression coefficients do not vary when the categories are changed and produce homogeneous estimate overall cut-off points to the effect of predictor variables on the outcome variable. Only the sign changes if the response categories are reversed. Both bivariable and multivariable ordinal logistic regression model was fitted to determine the association between the dependent and the independent variables and only factors that are significant at p-value less than 0.25 in bivariate analysis were entered into the multivariable analysis.

The overall good fit of the model was assessed using, the Likelihood ratio test (LRT) (Chi-square = 130.753, p-value <0.001) indicating a good fit of the model. Further, the proportionality assumption for the fitted proportional odds model was checked (LRT Chi-square = 28, p-value = 0.999), which confirms no violation of this assumption. Multi-collinearity was checked with the Variance inflation factor (VIF). There is no multicollinearity problem found (VIF < 10). Adjusted odds ratio with 95% CI, and p-value <0.05 was reported statistically significant.

## Ethical statement

Ethical review board of SPHMMC reviewed the study protocol, and approved ethical clearance (Ref No: Pm23/651). To reach health care professionals we have got letter of permission from head of SPHMMC research directorate. The right of participants to discontinue or refuse participation was respected and confidentiality was maintained. Verbal informed consent was taken in all participant before taking part in this study.

## Result

### Descriptive statistics

Out of 439 HCWs who participated in the study, 241(55%) were males. The mean age of the respondents was 31.23 years with a standard deviation of 6.23. Fifty percent of the respondents' (n = 220) were single, 206(46.9%) married, 9(2.1%) divorced. Concerning education level 13

**Table 2. Distribution of socio-demographic variables among HCWs of SPHMMC, Addis Ababa, Ethiopia,2023 (n = 439).**

| Variable | Category | %(n) |
|---|---|---|
| Sex | Male | 54.9 (241) |
|  | Female | 45.1 (198) |
| Age | 18–25 year | 10 (44) |
|  | 26–30 year | 49.4 (217) |
|  | 31–35 year | 20.3 (89) |
|  | 36–40 year | 13 (57) |
|  | 41–60 year | 7.3 (32) |
| Marital status | Single | 50.1(220) |
|  | Married | 46.9(206) |
|  | Divorced | 3(13) |
| Level of education | College Diploma | 3(13) |
|  | Bachler Degree | 53.8(236) |
|  | Master Degree | 14.4(63) |
|  | Doctorate Degree | 28.9(127) |
| Religion | Orthodox | 66.5(292) |
|  | Protestant | 21.2(93) |
|  | Muslim | 11.2(49) |
|  | Other | 1.1(5) |
| Monthly income | <8000birr | 46(202) |
|  | 8001–10,000 birr | 25.7(113) |
|  | 10001-14999birr | 18.9(83) |
|  | >15000birr | 9.3(41) |

(3%) had college diploma, 236(53.8%) had Bachler degree, 63(14.4%) had master degree, 127 (28.9%) had doctorate degree. Further, regarding religion, 292(66.5%) were Orthodox, 93 (21.2%) were protestant and 49(11.2%) were Muslim followers. Concerning monthly income 202(46%) of the respondents had earned less than eight thousand (8000) Ethiopian Birr or 141.72 United State Dollars (Table 2).

As the psychosocial factors were concerned twenty percent of the study participants faced childhood abuse, 51(11.6%) had a history of mental illness in their family, 36(8.2%) had a suicidal attempt and 212(48.3%) had a history of stressful life situations. Among those who sustain childhood abuse, 34(38.2%) were sustained by physical abuse, 26(29.2%) by verbal abuse, and 22(24.7%) by emotional abuse. Furthermore, among those who sustained stressful life events 25(45.5%) were due to the loss of a close family member or loved one, and 17(30.9%) were due to the loss of money or materials (Table 3).

Regarding physical status and social support, 45(10.3%) had sustained an injury, 11(2.5%) had a disability, and 42(9.6%) had a medical problem. Additionally, 146(33.3%) of study participants had headaches, 151(34.4%) had back pain, and 47(10.7%) had fever over the past 30 days respectively. Further, 31(7.1%) of the study participants had a smoking history. According to the Oslo 3-item social support scale, those who have poor, moderate, and strong social support constitute 156(35.5%), 223(50.8%), and 60(13.7%) of the study participants, respectively (Table 4).

Regarding the years of work experience, 191(43.5%) of study participants had 2–5 years of experience, and 146(33.3%) had 10–15 years of experience. Regarding the working department and environment, 88(20%) work in the medical department, 79(18.3%) work in the surgical

**Table 3. Distribution of psychosocial factors among HCWs of SPHMMC,2023 (n = 439).**

| Variable | Category | % (n) |
|---|---|---|
| History of childhood abuse | Yes | 20.5(90) |
| | No | 79.5(349) |
| Presence of family history of mental illness | Yes | 11.6(51) |
| | No | 88.4(388) |
| History of suicidal attempt | Yes | 8.2(36) |
| | No | 91.8(403) |
| History of stressful life events | Yes | 48.3(212) |
| | No | 51.7(227) |

department, 13 (3%) in the ophthalmic, and 11(2.5%) in the ear, nose and throat department. Further, 95(21.6%) of respondents working environment was emergency, and 190(43.3%) work as inpatient. Moreover, regarding working hours and level of job satisfaction, 278 (63.3%) work more than 8 hours, and 207(47%) of respondents are currently dissatisfied with their job (Table 5).

## Prevalence of depression among health care workers at SPHMMC

Based on the cut of point $\geq 10$ for cases on PHQ-9, the prevalence of depression among HCWs in SPHMMC was found 96 (21.9% (95%CI: 18%, 27.76%)), of which 56 (58.3%) were male. Concerning the severity, 41(9.3%) had severe depression, 55(12.5%) had moderate depression,

**Table 4. Distribution of physical health status and social support among HCWs of SPHMMC, 2023(n = 439).**

| Variable | Category | %(n) |
|---|---|---|
| History of Injury | Yes | 10.3(45) |
| | No | 89.7(394) |
| Presence of disability | Yes | 2.5(11) |
| | No | 97.5 (428) |
| Presence of headache over the past 30 days | Yes | 33.3(146) |
| | No | 66.7(293) |
| Presence of back pain over the past 30 days | Yes | 34.4(151) |
| | No | 65.6(288) |
| Presence of fever over the past 30 days | Yes | 10.7(47) |
| | No | 89.3(392) |
| Presence of medical problem | Yes | 9.6(42) |
| | No | 90.4(397) |
| Smoking history | Yes | 7.1(31) |
| | No | 92.9(408) |
| Social support | Poor | 35.5(156) |
| | Moderate | 50.8(223) |
| | Strong | 13.7(60) |

Almost half of the study participants were nurses. There were 90(20.5%) medical specialties including residency, and 35(8%) general practitioners. In terms of other professions, 23(5.2%) were midwives, 19(4.3%) were medical laboratories, 17(3.9%) were public health, 15(3.4%) were pharmacists, 10(2.3%) were anesthesiologists, 8(1.8%) were radiologists, and 15(3.4%) were others such as Psychiatry professionals, Phlebotomist, Biomedical engineer, and Biomedical technician.

**Table 5. Distribution of professional factor, job satisfaction and work area factor among HCWs of SPHMMC,2023(n = 439).**

| Variable | Category | %(n) |
|---|---|---|
| Field of study | Nursing | 47.2(207) |
| | Midwifery | 5.2(23) |
| | Medical doctor | 8(35) |
| | Specialty | 20.5(90) |
| | Medical laboratory | 4.3(19) |
| | Pharmacy | 3.4(15) |
| | Anesthesia | 2.3(10) |
| | Public health | 3.9(17) |
| | Other | 3.4(15) |
| | Radiology | 1.8(8) |
| Working hour | More than 8 hours | 63.3(278) |
| | 8 hours | 31.4(138) |
| | Less than 8 hours | 5.2(23) |
| Work experience | Less than 2 years | 7.5(33) |
| | 2–5 year | 43.5(191) |
| | 5–10 year | 33.3(146) |
| | 10–15 year | 10.9(48) |
| | Above 15 years | 4.8(21) |
| Working department | Surgical | 18(79) |
| | Medical | 20(88) |
| | Pediatrics | 15.3(67) |
| | Obstetrics | 16.9(74) |
| | Trauma and burn center | 7.1(31) |
| | Laboratory | 3.9(17) |
| | Pharmacy | 3.4(15) |
| | Ophthalmology | 3(13) |
| | ENT | 2.5(11) |
| | Radiology | 3.4(15) |
| | Psychiatry | 2.1(9) |
| | Other | 2.1(9) |
| | Anesthesia | 2.5(11) |
| Work Environment | Emergency | 21.6(95) |
| | Outpatient | 24.6(108) |
| | Inpatient | 43.3(190) |
| | Radiology | 3.4(15) |
| | Office | 7.1(31) |
| Job satisfaction | Dissatisfied | 47(207) |
| | Satisfied | 53(232) |

and 152(34.6%) had mild depression (Fig 2). (Fig 2. Prevalence of Depression by Severity Among HCWs, in SPHMMC, Addis Ababa, Ethiopia, 2023).

## Factors associated with depression among health care workers at SPHMMC

In model fitting, we started with all candidate predictors significant at 25% in the bivariate analysis and then employed backward elimination methods to reach the final model. This approach is useful because it reduces the number of predictors, reducing the multicollinearity

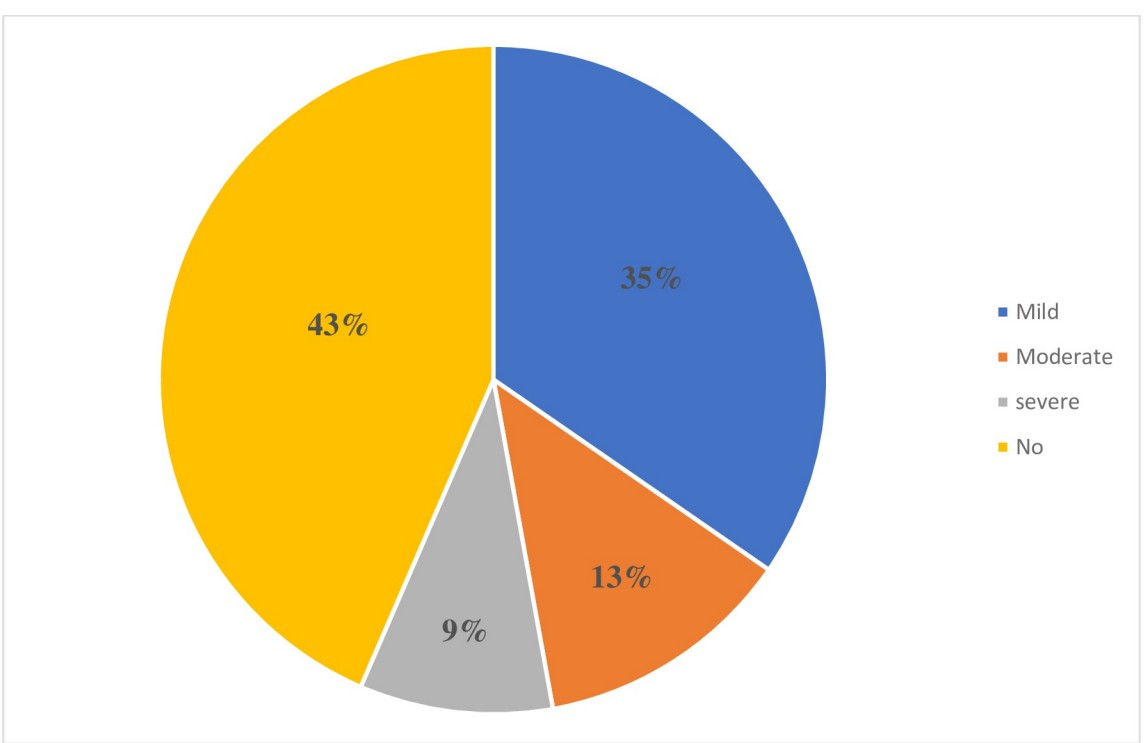

**Fig 2. Prevalence of depression by severity among HCWs, in SPHMMC, Addis Ababa, Ethiopia, 2023.**

problem and it is one of the ways to resolve the overfitting. The overall good fit of the final model was assessed using, the LRT of the goodness-of-fit test (Chi-square = 130.753) with p-value <0.001 indicating a good fit of the model. Having fitted the proportional odds model, the proportionality assumption for the fitted proportional odds model was checked using the LRT, which confirms no violation of this assumption (Chi-square = 28, p-value = 0.999). Multi-collinearity was checked with VIF. There is no multicollinearity problem found.

In the multivariable analysis, being single (AOR = 7.78, 95% CI (1.23, 49.011) was found to be highly associated with increased odds of depression than the divorced state. This wider interval is mainly related to the small sample size for the divorce category. Further, sustained childhood abuse (AOR = 2.57, 95% CI (1.49, 4.42)), history of suicidal attempts (AOR = 2.66, 95% CI (1.25, 5.67)), and history of stressful life events (AOR = 1.527, 95% CI (1.02, 2.3)) were associated with a greater odd of depression, compared to their counterpart did not sustain childhood abuse, no history of suicidal attempt and no history of stressful life event, respectively.

Moreover, a multivariable analysis revealed that individuals who had experienced back pain in the previous 30 days had a higher risk of depression than those who had not (AOR = 2, 95% CI (1.3, 3.11)). In a similar vein, HCWs who worked longer shifts than eight hours (AOR = 3.03, 95% CI (1.119, 8.245)) had increased odds of depression than those who worked shorter shifts. HCWs with work experience of between five and ten years (AOR = 4, 95% CI (1.05, 15.27)) and between ten and fifteen years (AOR = 4.24, 95% CI (1.08, 16.58)) were more likely to experience increased odd of depression than those with work experience of more than fifteen years, and those with poor social support (AOR = 2.09(1.09, 3.99)) were more likely to experience increased odds of depression than those with strong social support (Table 6).

**Table 6. Multivariable analysis of different variables and depression among HCWs of SPHMMC Addis Ababa, Ethiopia,2023(n = 439).**

| Variable | Category | Depression | | | | P-Value | COR (95%CI) | P-Value | AOR (95%CI) |
|---|---|---|---|---|---|---|---|---|---|
| | | Severe%(n) | Moderate %(n) | Mild %(n) | No %(n) | | | | |
| Age | 18–25 year | 5(2) | 7(3) | 43(19) | 45(20) | 0.20 | 1.795 (0.727,4.430) | 0.41 | 0.55(0.13,2.29) |
| | 26–30 year | 14(30) | 14(31) | 35(76) | 37(80) | 0.003** | 3.16(1.465,6.818) | 0.96 | 0.97(0.310,3.05) |
| | 31–35 year | 6(5) | 15(13) | 36 (32) | 44(39) | 0.06 | 2.16(0.951,4.911) | 0.47 | 0.66(0.22,1.99) |
| | 36–40 year | 4(2) | 9(5) | 33(19) | 54(31) | 0.46 | 1.39(0.579,3.353) | 0.19 | 0.47(0.156,1.442) |
| | 41–60 year | 6(2) | 9(3) | 19(6) | 66(21) | | 1.00 | | 1.00 |
| Marital status | Single | 11(24) | 12(27) | 38(83) | 39(86) | 0.04* | 5.02 (1.025,24.558) | 0.03* | 7.78(1.23,49.011) |
| | Married | 8(17) | 13(26) | 32(66) | 47(97) | 0.09 | 3.82 (0.779,18.733) | 0.04* | 6.7(1.103,40.75) |
| | Divorced | 0% | 15(2) | 23(3) | 62(8) | | 1.00 | | 1.00 |
| Level of education | College Diploma | 0% | 0% | 15(2) | 85(11) | 0.04* | 0.19(0.042,0.900) | 0.16 | 0.197(0.021,1.85) |
| | Bachler Degree | 11(27) | 12(29) | 36(86) | 40(94) | 0.01** | 1.7(1.130,2.563) | 0.36 | 1.89(0.48,7.49) |
| | Master Degree | 10(6) | 22(14) | 38(24) | 30(19) | 0.002** | 2.43(1.39,4.236) | 0.29 | 2.17(0.517,9.129) |
| | Doctorate Degree | 6(8) | 9(12) | 31 (40) | 53 (67) | | 1.00 | | 1.00 |
| Monthly income in ETB | <8000 | 12(25) | 14(28) | 33(67) | 41(82) | 0.002** | 2.87(1.478,5.634) | 0.205 | 1.75(0.738,4.131) |
| | 8,001–10,000 | 8(9) | 14(16) | 38(43) | 40(45) | 0.007** | 2.65(1.312,5.351) | 0.154 | 1.83(0.798,4.184) |
| | 10,001–14,999 | 8(7) | 10(8) | 36(30) | 46(38) | 0.04* | 2.13(1.020,4.431) | 0.234 | 1.64(0.726,3.708) |
| | >15,000 | 0% | 7(3) | 29(12) | 63(26) | | 1.00 | | 1.00 |
| History of childhood abuse | Yes | 16(14) | 20(18) | 32(36) | 26(29) | 0.001* | 2.3(1.497,3.529) | 0.001** | 2.57(1.49,4.42) |
| | No | 8(27) | 11(37) | 34 (120) | 47 (165) | | 1.00 | | 1.00 |
| Presence of family history of mental illness | Yes | 12(6) | 12(6) | 43(22) | 33(17) | 0.20 | 1.41(0.833,2.388) | 0.06 | 0.535(0.28,1.025) |
| | No | 9(35) | 13(49) | 34 (130) | 45 (174) | | 1.00 | | 1.00 |
| History of suicidal attempt | Yes | 22(8) | 19(7) | 36(13) | 8(22) | 0.001** | 2.94(1.572,5.485) | 0.011* | 2.66(1.25,5.67) |
| | No | 8(33) | 12(48) | 34 (139) | 45 (183) | | 1.00 | | 1.00 |
| History of stressful life events | Yes | 12(25) | 16(34) | 40(84) | 33(69) | 0.001** | 2.24(1.570,3.189) | 0.041* | 1.527(1.02,2.3) |
| | No | 7(16) | 9(21) | 30(68) | 54 (122) | | 1.00 | | 1.00 |
| Presence of headache over the past 30 days | Yes | 12(18) | 19(28) | 40(59) | 28(41) | 0.001** | 2.42(1.672,3.505) | 0.161 | 1.384(0.88,2.18) |
| | No | 8(23) | 9(27) | 32(93) | 51 (150) | | 1.00 | | 1.00 |
| Presence of back pain over the past 30 days | Yes | 12(18) | 18(27) | 44(67) | 26(39) | 0.001** | 2.56(1.772,3.691) | 0.002** | 2 (1.30,3.11) |
| | No | 8(23) | 10(28) | 30(85) | 53 (152) | | 1.00 | | 1.00 |
| Presence of fever over the past 30 days | Yes | 17(8) | 15(7) | 40(19) | 28(13) | 0.01** | 2.03(1.169,3.514) | 0.69 | 1.133(0.612,2.09) |
| | No | 8(33) | 12(48) | 8(33) | 45 (178) | | 1.00 | | 1.00 |
| Presence of medical problem | Yes | 17(7) | 10(4) | 38(16) | 36(15) | 0.22 | 1.45(0.805,2.607) | 0.370 | 1.38(0.68,2.76) |
| | No | 9(34) | 13(51) | 34 (136) | 44 (176) | | 1.00 | | 1.00 |
| Field of study | Nursing | 10(21) | 15(31) | 35(73) | 40(82) | 0.41 | 1.73(0.468,6.413) | 0.099 | 0.137 (0.013,1.451) |
| | Medical doctor | 6(8) | 10(12) | 33(41) | 51(64) | 0.37 | 1.92(0.467,7.859) | 0.239 | 0.199 (0.014,2.926) |
| | Other | 12(12) | 11(11) | 35(35) | 41(41) | 0.23 | 2.55 (0.561,11.605) | 0.087 | 0.121 (0.011,1.361) |
| | Radiology | 0 | 13(1) | 38(3) | 50(4) | | 1.00 | | 1.00 |

(*Continued*)

**Table 6.** (Continued)

| Variable | Category | Depression | | | | P-Value | COR (95%CI) | P-Value | AOR (95%CI) |
|---|---|---|---|---|---|---|---|---|---|
| | | Severe%(n) | Moderate %(n) | Mild %(n) | No %(n) | | | | |
| Working hour | More than 8 hours | 10(28) | 15(43) | 37 (104) | 37 (103) | 0.030* | 2.55(1.093,5.941) | 0.029* | 3.03(1.119,8.245) |
| | 8 hours | 9(12) | 7(10) | 30(42) | 54(74) | 0.50 | 1.36(0.564,3.268) | 0.196 | 1.97(0.70,5.51) |
| | Less than 8 hours | 4(1) | 9(2) | 26(6) | 61(14) | | 1.00 | | 1.00 |
| Work experience | Less than 2 years | 3(1) | 12(4) | 45(15) | 39(13) | 0.036* | 3.29(1.082,9.992) | 0.085 | 4.32(0.82,22.819) |
| | 2–5 year | 13(25) | 10(20) | 36(69) | 40(77) | 0.005** | 3.96 (1.501,10.419) | 0.06 | 3.83(0.952,15.38) |
| | 5–10 year | 10(15) | 14(21) | 34(49) | 42(61) | 0.008** | 3.79 (1.423,10.092) | 0.04* | 4 (1.05,15.27) |
| | 10–15 year | 0 | 19(9) | 29(14) | 52(25) | 0.11 | 2.41(0.822,7.044) | 0.038* | 4.24(1.08,16.58) |
| | Above 15 years | 0 | 5(1) | 24(5) | 71(15) | | 1.00 | | 1.00 |
| Working department | Major | 10(33) | 13(44) | 35 (117) | 43 (145) | 0.581 | 0.746 (0.264,2.109) | 0.402 | 0.571 (0.154,2.118) |
| | Minor | 8(7) | 11(10) | 33(29) | 48(43) | 0.354 | 0.596(0.2,1.778) | 0.494 | 0.611 (0.152,2.478) |
| | Anesthesiology | 9(1) | 9(1) | 55(6) | 27(3) | | 1.00 | | 1.00 |
| Work Environment | Emergency | 12(11) | 17(16) | 42(40) | 29(28) | 0.027* | 2.41(1.104,5.256) | 0.45 | 1.43(0.57,3.57) |
| | Outpatient | 9(10) | 14(15) | 26(28) | 51(55) | 0.59 | 1.24(0.567,2.699) | 0.75 | 1.16(0.47,2.85) |
| | Inpatient | 8(16) | 11(21) | 37(71) | 43(82) | 0.33 | 1.45(0.691,3.026) | 0.84 | 1.09(0.45,2.68) |
| | Radiology | 0 | 13(2) | 27(4) | 60(9) | 0.65 | 0.76(0.224,2.543) | 0.28 | 0.33(0.044,2.445) |
| | Office | 13(4) | 3(1) | 29(9) | 55(17) | | 1.00 | | 1.00 |
| Job satisfaction | Dissatisfied | 10(21) | 15(31) | 39(80) | 36(75) | 0.007** | 1.62(1.14,2.30) | 0.121 | 1.36(0.92, 2) |
| | Satisfied | 9(20) | 10(24) | 31(72) | 50 (116) | | 1.00 | | |
| Social support | Poor | 13(21) | 15(23) | 38(59) | 34(53) | 0.001** | 2.59(1.474,4.562) | 0.025* | 2.09(1.09,3.99) |
| | Moderate | 8(17) | 13(30) | 32(71) | 47 (105) | 0.111 | 1.56(0.904,2.676) | 0.35 | 1.35(0.73,2.49) |
| | Strong | 5(3) | 3(2) | 37(22) | 55(33) | | 1.00 | | 1.00 |

NB: * = significant at 0.05, ** = significant at 0.01 COR = Crude odd ratio AOR = Adjusted odd ratio CI = Confidence interval n = frequency

## Discussion

This study assessed the prevalence and the factors associated with depression among HCWs. The overall prevalence of depression in this study was 21.9% (95%CI: 18%, 27.76%); while the prevalence of mild, moderate, and severe depression was 35%, 13%, and 9%, respectively. This is nearly agreed with research conducted in Eka Kotebe Hospital 23.5% [28], Dessie Comprehensive Specialized Hospital 27.8% [29], and Gurage zonal public hospitals 25.8% [30].

However, the proportion of depression, in our study was higher when compared to studies conducted in Nigeria 9.8% [31], and 14.9% [32], Uganda 12.4% [33], and Kenya 15.4% [34]. This discrepancy could be the result of different study designs and instruments; for example, one study conducted in Nigeria used the Zung self-rating depression scale, whereas this study used a PHQ-9. Further differences in sample size and professional mix may be other reasons for the discrepancy. The study done in Nigeria comprised 1452 individuals, all of whom were only doctors and nurses, and our study included participants from a variety of health professions. Furthermore, another potential explanation could be the variation in participants' socioeconomic and demographic features within the communities.

On the other hand, the findings from this study were lower compared to studies done in Saudi Arabia 43.9% [35], Southern China public hospital 38% [36], Southern Ethiopia 50.1% [21], Gondar comprehensive specialized Hospital 42.7% [22], Eastern Ethiopia 66.4% [19], central Ethiopia 60.3% [20], China 57.2% [37], Kenya 53.6% [7], Baghdad 70.25% [38], Egypt 71.4% [39] and Sudan 75% [16]. The discrepancy might be due to variation in study participants; for instance, a study conducted in a southern China public hospital included 3474 study participants all of whom were nurses while in this study 439 study participants from different professions were included.

Another potential reason for this discrepancy might be due to time variation. A study done in the three public hospitals of Kenya [7], Gondar Comprehensive Specialized Hospital [22], Sudan [16], Southern Ethiopia [21], Eastern Ethiopia [19], and Central Ethiopia 60.3 [20] was conducted during the era of COVID-19 while this study was conducted post COVID-19 pandemic. In addition to this working environment and probably the difference in a medical setting might be the possible reason for the difference in prevalence of depression among HCWs.

The factors associated with depression in this study were having a history of abuse as a child, having attempted suicide in the past, having back pain in the past, being single, working more than eight hours a day, having more years of work experience, and having poor social support. The study found that the odds of having depression are 7.78 times higher among those HCWs who were single as compared to those who were divorced. This result was in line with previous results from Eka Kotebe Hospital [28], Dessie Comprehensive Specialized Hospital [29], and Southern Ethiopia [21]. Being single was the psychosocial risk factor of depression due to a lack of a partner, thereby lacking social support when in need of it.

In this study, the presence of childhood abuse among HCWs was 2.57 times at a higher level of depression compared to no history of childhood abuse. A systematic review and meta-analysis study also shows child maltreatment had 1.07 higher odds of having depression [40, 41]. The fact that the effect of childhood abuse in any form leads to depression even in chronic form is demonstrated by other studies that have been connected to changes in a person's brain structure. Moreover, it can significantly raise the likelihood of adult depression and suicide [40–42].

Further, working more than 8 hours per day and having work experience of 5–15 years were associated with depression. Working more than 8 hours per day and having work experience of 5–15 years are 3 and 4 times at high risk of developing depression respectively. This is in line with studies done in Japan [43] and China [44]. Long working hour leads to depression because those who spend more of their time at work do not have enough relaxation time and those who have more years of experience have more responsibility in patient care which leads to a higher risk of depression.

Poor social support is another factor found to be significantly associated with depression among HCWs. Those who had poor social support had a 2 times higher risk of developing depression than those who had strong social support. This is agreed with studies done in Ethiopia [45], Brazil [46], and China [47]. This is because those who had poor social support have no friends or relatives whom they can count on in times of trouble which predisposes them to depression.

Furthermore, this study found that those who had a history of suicidal attempts and stressful life events ever present were significantly associated with depression. Suicidal attempts and stressful life events ever present were 2.6 and 1.52 times at high risk of developing depression respectively. This was also stated by different studies from Australia and Malaysia, in which the odds of depression increase by 1.24 among those who attempt suicidal in Australia and 55% among those who have suicidal ideation in Malaysia [48, 49]. This is because there is a substantial correlation between depression and suicidal thoughts or attempts, and HCWs who

are depressed may exhibit these symptoms. Moreover, HCWs with a history of stressful life events are more vulnerable to real or perceived threats, which can lead to the development of glucocorticoid resistance along with excessive inflammation, which raises the risk of development of depression [50].

In the current study, back pain was another factor associated with depression among HCWs. Those HCWs who had back pain over the past 30 days were 2 times more likely to experience depressive symptoms. This result was nearly agreed by a previous studies done in Turkey in which back pain increase depression by 61.4% [51]. This could be because persistent back pain causes neuronal plasticity, which directly contributes to depression, and because chronic pain raises stress levels and sleep deprivation, both of which contribute to the occurrence of depression [52].

## Limitations of the study

Determining causality was not possible with a cross-sectional study design. The fact that this study was limited to one institution makes it impossible to share a wealth of data regarding the prevalence of depression among HCWs employed by other institutions. This study's inability to assess chronic depression, taking yes or no response to assess suicidal attempt and chronic disease status was another drawback. Furthermore, an in-depth understanding of depression could have been attained by qualitative or mixed-method studies, but the quantitative nature of this study made this impossible.

## Conclusion

In this research, we looked at the prevalence of depression and the factors associated with it among HCWs at SPHMMC. The outcome demonstrates that HCWs have a higher prevalence of depression than does general population [1]. HCWs with a history of suicidal attempts, those who had a history of stressful life events, back-pain stories, single marital status, history of childhood abuse, more than eight hours of work per day, more years of work experience, and poor social support were significantly associated with depression. Therefore, this HCWs should receive special attention by SPHMMC for early screening and treatment of depression.

## Supporting information

**S1 Data.**
(SAV)

## Acknowledgments

We would like to thank the public health department at SPHMMC for supporting this research. We extend our heartfelt gratitude to the higher authorities at SPHMMC for their dedicated cooperation. We also thank all research participants for their unwavering commitment.

## Author Contributions

**Conceptualization:** Melese Bahiru Tesema, Berhanu Teshome Woldeamanuel, Eyoel Berhane Mekonen.

**Data curation:** Melese Bahiru Tesema, Berhanu Teshome Woldeamanuel, Eyoel Berhane Mekonen, Kidest Getu Melese.

**Formal analysis:** Melese Bahiru Tesema, Berhanu Teshome Woldeamanuel.

**Investigation:** Melese Bahiru Tesema, Berhanu Teshome Woldeamanuel, Eyoel Berhane Mekonen.

**Methodology:** Melese Bahiru Tesema, Berhanu Teshome Woldeamanuel, Kidest Getu Melese.

**Supervision:** Melese Bahiru Tesema, Berhanu Teshome Woldeamanuel, Eyoel Berhane Mekonen, Kidest Getu Melese.

**Visualization:** Melese Bahiru Tesema, Kidest Getu Melese.

**Writing – original draft:** Melese Bahiru Tesema, Berhanu Teshome Woldeamanuel.

**Writing – review & editing:** Melese Bahiru Tesema, Berhanu Teshome Woldeamanuel, Eyoel Berhane Mekonen.

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
