## [Decision Letter · Decision Letter 0]

24 Apr 2024

PONE-D-24-07891Depressive Symptoms and its Associated Factors among Health Care Workers in Saint Paul’s Hospital Millennium Medical College, Ethiopia.PLOS ONE

Dear Dr. Tesema

Thank you for submitting your manuscript to PLOS ONE. After careful consideration, we feel that it has merit but does not fully meet PLOS ONE’s publication criteria as it currently stands. Therefore, we invite you to submit a revised version of the manuscript that addresses the points raised during the review process.

We look forward to receiving your revised manuscript.

Kind regards,

Chalachew Kassaw Demoze, Msc

Academic Editor

PLOS ONE

Journal Requirements:

Additional Editor Comments:

Dear reviwer, We would like to acknowledge your work.

Hereunder, I suggest some corrections to your manuscript.

1. Your study topic is extensively researched, so what makes your tudy unique, novel and relevant?

2. Why do you want to do ordinal rather than dichotomous or logistic regression analysis?

3. it is better to write an assumption of ordinal regression analysis

4. It is better to write the data collection tool descriptions you used to assess independent variables.

5. it is better to write concrete reasons for the study result difference ( "the inclusion criteria, sample size and instrument used " reasons might not be statistically acceptable).

6. it is better to write a specific recommendation for each stduy finding to the institution (St. Paul paul college)

Reviewers' comments:

Reviewer's Responses to Questions

**Comments to the Author**

1. Is the manuscript technically sound, and do the data support the conclusions?

Reviewer #1: Partly

Reviewer #2: Yes

2. Has the statistical analysis been performed appropriately and rigorously? 

Reviewer #1: Yes

Reviewer #2: Yes

3. Have the authors made all data underlying the findings in their manuscript fully available?

Reviewer #1: Yes

Reviewer #2: No

4. Is the manuscript presented in an intelligible fashion and written in standard English?

Reviewer #1: Yes

Reviewer #2: No

5. Review Comments to the Author

Reviewer #1: The authors raised important topic but need to solidify their study on research gaps and the need of this study and why they need to do it

Abstract:

Background: It is better to describe the burden of depression overall and targeting the health care worker rather than putting the definition

Conclusion: Do you think this recommendation is feasible and how can MOH know this about healthcare workers and we all know stigma faced people with mental illness.

Background

Paragraph 2 needs synthesis and shortening also you could consider making it two paragraphs

Need to work on literature gap and novelty

what makes your study different if already done in Ethiopia

what are the research gaps and please back it up with evidences

methodology

What is the reason using power calculation each factor?

it’s better to describe the factors and dependent variable in separate heading

describe the value of multicoliniarity and model fitness the exact statically number

Results

How come you used cut of point of 5 as depression as the score depression is not like that but showing the severity that made the prevalence higher

please show the p value in the table

the discussion needs to be synthesize the argument and supporting literature

More to discuss the reason behind it

you need to re-run the prevalence i don't think there is higher prevalence because so you used 5 as cut point because you include mild

Reviewer #2: - The authors refers to Depressive disorder, but also Depression, Depressive symptoms, Depressed symptoms, Major depressive disorder etc… interchangeably, which can be confusing to the reader. These phenomena are not the same thing, and it is advisable to use one term consistently throughout the abstract and main text. In this case, either Depression or Depressive symptoms.

- In Paragraph 3 of the background - it is important to clarify that those studies were also done among HCW. Only the first sentence mentions HCW.

- The last paragraph of the background is too repetitive. The aim of study only needs to be stated once.

- In the sample size section, it is not clear how the authors came up with this method of sample size calculation. If this is a pre-existing method then a citation needs to be added, otherwise if this is an entirely new method of size calculation that also needs to be stated.

- In the Statistic Analysis, name should be Microsoft Excel, and mention the version used

- For the results, the authors started very well using past tense (were, was) but then later on switched to present tense (are, is, has). When writing results and discussion, the standard rule is to use past tense because we are mentioning what was found at the time of data collection, not now.

- Decide whether you want to use the name in full, or abbreviation SPHMMC, but not both

- What do the authors think about HCW responding to questions about suicide attempts, and chronic health issues? Could this be limitation? How did they address it?

6. PLOS authors have the option to publish the peer review history of their article (what does this mean?). If published, this will include your full peer review and any attached files.

Reviewer #1: No

Reviewer #2: **Yes: **Kim Madundo

---

## [Author Response · Author response to Decision Letter 0]

31 May 2024

Response to reviewers

Reviewer #1: 

1. The authors raised important topic but need to solidify their study on research gaps and the need of this study and why they need to do it

Response: Health care workers working in general set-up (non-front line) are susceptible to mental illness specifically depression. But research done among health care worker on depression among those whose working condition as non-frontline (general health care setup) is limited. So, this study helps to fill the above research gap by studying prevalence and factor associated with depression among health care workers in SPHMMC. 

2. Abstract:

Background: It is better to describe the burden of depression overall and targeting the health care worker rather than putting the definition

Response: We did a revision accordingly (see Abstract, lines 2-5)

Conclusion: Do you think this recommendation is feasible and how can MOH know this about healthcare workers and we all know stigma faced people with mental illness.

Response: we added descriptions accordingly (Abstract lines 25-28)

3. Background

Paragraph 2 needs synthesis and shortening also you could consider making it two paragraphs

Response: we had split this paragraph into two (background paragraph 2-3)

4. Need to work on literature gap and novelty

Response: we added a reason for literature gabs (page 2, last paragraph of background line 12-17)

5. what makes your study different if already done in Ethiopia

Response: the available literature in Ethiopia mainly focused on depression among front line HCWs during COVID-19.

6. what are the research gaps and please back it up with evidences

Response: As stated on page 2, lines 12-17, the most recent available literature in Ethiopia mainly focused on depression among front line HCWs during COVID-19. Research done among HCWs working in general set up is limited in Ethiopia as well as in study area. This shows that there is literature gap in the area related to HCWs working as non-front line. (see reference listed 19-22)

7. Methodology

What is the reason using power calculation each factor?

Response: Sample size calculations are used to determine either the sample size required to give a specified power, or the power implied by a specified sample size. Thus, sample size calculations need to be undertaken prior to a study to avoid both the wasteful consequences of under-powering or of overpowering. Further, in this study we planned to perform a test of hypothesis comparing the proportions, with an ordered categorical outcome, where analysis is by the proportional odds model. Studies that use small samples not being able to demonstrate differences or associations which really do exist. Such studies are ‘under-powered’, not possessing sufficient statistical power to detect the effects they set out to detect. The statistical power of a study builds in a safety margin to avoid generalizing false positive results which could have expensive consequences. A non-significant finding of a study may thus simply reflect the inadequate power of the study to detect differences or associations at levels which are conventionally accepted as statistically significant. In this study, the target power is 80%, meaning that a study has an 80% likelihood of detecting an association which really exists.

8. it’s better to describe the factors and dependent variable in separate heading

Response: We added descriptions for both dependent and independent factors separately. (methods, page 4 lines 5-116 and page 5 lines 1-5)

9. describe the value of multicollinearity and model fitness the exact statically number

Response: We had added description about the cutoff points to multicollinearity test and overall goodness of fit test. (Statistical analysis: page 6 lines 16-22)

10. Results

How come you used cut of point of 5 as depression as the score depression is not like that but showing the severity that made the prevalence higher

Response: Based on recommendation, we used cutoff point of ≥10 to show the prevalence of depression. Using cutoff point ≥10 for cases on PHQ-9, the prevalence of depression among HCWs in SPHMMC becomes 21. 9%, (95%CI: 18% ,27.76%).

11. Please show the p value in the table

Response: Confidence intervals are preferable to p-values, as they tell us the range of possible effect sizes compatible with the data. Further, confidence interval that excludes 1 reveals significant association. P-values simply provide a cut-off beyond which we assert that the findings are 'statistically significant. Thus, we used notation to show the strength of association to those significant variables (see bottom of Table 6). Also, we added p-values in this revised version.

12. the discussion needs to be synthesized the argument and supporting literature. More to discuss the reason behind it

Response: we did edition and added a reason in the revised version accordingly.

13. you need to re-run the prevalence i don't think there is higher prevalence because so you used 5 as cut point because you include mild.

Response: We used the classification as severe depression, moderate depression, and mild depression only to show the level of variation in the depression status. This is to show the variability as we move from lower level to higher level depression status. But based on the cutoff point ≥10 for cases on PHQ-9, the prevalence of depression among HCWs in SPHMMC was found 21. 9%, (95%CI: 18% ,27.76%). Concerning the severity, 9.3% had severe depression, and 12.5% had moderate depression, while 34.6% had mild depression

Reviewer #2: 

1. The authors refer to Depressive disorder, but also Depression, Depressive symptoms, Depressed symptoms, Major depressive disorder etc… interchangeably, which can be confusing to the reader. These phenomena are not the same thing, and it is advisable to use one term consistently throughout the abstract and main text. In this case, either Depression or Depressive symptoms.

Response: It is depression, we did corrections throughout the document.

2. In Paragraph 3 of the background - it is important to clarify that those studies were also done among HCW. Only the first sentence mentions HCW. 

Response: We added description on paragraph 4-line number 5 and 8 as if those study were done among HCWs.

3. The last paragraph of the background is too repetitive. The aim of study only needs to be stated once. 

Response: we did corrections in the revised version

4. In the sample size section, it is not clear how the authors came up with this method of sample size calculation. If this is a pre-existing method then a citation needs to be added, otherwise if this is an entirely new method of sample size calculation that also needs to be stated. 

Response: Sample size calculations are used to determine either the sample size required to give a specified power, or the power implied by a specified sample size. Thus, sample size calculations need to be undertaken prior to a study to avoid both the wasteful consequences of under-powering or of overpowering. Further, in this study we planned to perform a test of hypothesis comparing the proportions, with an ordered categorical outcome, where analysis is by the proportional odds model. Studies that use small samples not being able to demonstrate differences or associations which really do exist. Such studies are ‘under-powered’, not possessing sufficient statistical power to detect the effects they set out to detect. The statistical power of a study builds in a safety margin to avoid generalizing false positive results which could have expensive consequences. A non-significant finding of a study may thus simply reflect the inadequate power of the study to detect differences or associations at levels which are conventionally accepted as statistically significant. In this study, the target power is 80%, meaning that a study has an 80% likelihood of detecting an association which really exists. (See Whitehead J. Sample size calculations for ordered categorical data. Stat Med. 1993 Dec 30;12(24):2257-71. Erratum in: Stat Med 1994 Apr 30;13(8):871. PMID: 8134732).

5. In the Statistic Analysis, name should be Microsoft Excel, and mention the version used

Response: it is Microsoft Excel version 19 

6. For the results, the authors started very well using past tense (were, was) but then later on switched to present tense (are, is, has). When writing results and discussion, the standard rule is to use past tense because we are mentioning what was found at the time of data collection, not now. 

Response: We did extensive language edition in the revised version.

7. Decide whether you want to use the name in full, or abbreviation SPHMMC, but not both

Response: We had corrected in the revised version

8. What do the authors think about HCW responding to questions about suicide attempts, and chronic health issues? Could this be limitation? How did they address it?

Response: Yes, it could be limitation and we added it under limitation of study

Additional Editor Comments:

1. Your study topic is extensively researched, so what makes your study unique, novel and relevant?

Response: What makes this study novel, unique and relevant is study being conducted among HCWs working in general working setup (in all working department) rather than frontline HCWs. Further, the previous studies consider to model only the presence of depression as yes or no. Even if the individual has depression the level of severity also varies among individuals. Thus, this study tries to fill this research gab. 

2. Why do you want to do ordinal rather than dichotomous or logistic regression analysis?

Response: we added a reason for the use of ordinal logistic regression model in the revised manuscript (page 6, lines 3-11 and page 6, lines 16-22)

3. it is better to write an assumption of ordinal regression analysis

Response: we added the assumptions of our model in the revised revision of the manuscript (page 6, lines 8-11)

4. It is better to write the data collection tool descriptions you used to assess independent variables.

Response: Sociodemographic variables, psycho-social factors, physical health status, and health professional status were assessed by HCWs response to yes/no options and among listed category for each question. To assess social support Oslo-3-item social support scale was used and those with score of 3-8 was classified as poor social support, 9-11 as moderate social support and 12-14 as strong social support. Page 5, lines 11-15. (See Kocalevent RD, Berg L, Beutel ME, Hinz A, Zenger M, Härter M et al. Social support in the general population : standardization of the Oslo social support scale ( OSSS-3 ). BMC Psychology. 2018;6(31):1–8.) Job satisfaction was measured by using short form twenty items Minnesota Satisfaction Questionnaire each scored 5-point Likert scale with 1 denoting strongly dissatisfied and 5 denoting strongly satisfied. Overall job satisfaction was estimated by taking the sum score of all the subscales. HCWs who scored a value of 62 and below was considered as dissatisfied and those who scored greater than 62 was considered as satisfied (Reference_ Weiss J. David, Rene D V. Manual for the Minnesota Satisfaction Questionnaire.pdf. 1967. p. 110–1). This description is added in revised version page 5-line number 15-20.

5. it is better to write concrete reasons for the study result difference (“the inclusion criteria, sample size and instrument used " reasons might not be statistically acceptable).

Response: correction done in the revised version

6. it is better to write a specific recommendation for each study finding to the institution (St. Paul college)

Response: We added recommendation to the institution (SPHMMC) in the abstract as well as conclusion part.

NB. We added reference and cited accordingly. As a result, our reference list increased from 46 to 52.

---

## [Decision Letter · Decision Letter 1]

6 Aug 2024

Depression and its Associated Factors among Health Care Workers in Saint Paul’s Hospital Millennium Medical College, Ethiopia.

PONE-D-24-07891R1

Dear Authors;

We’re pleased to inform you that your manuscript has been judged scientifically suitable for publication and will be formally accepted for publication once it meets all outstanding technical requirements.

Kind regards,

Shegaw Tesfa Mengist

Academic Editor

PLOS ONE

Additional Editor Comments (optional):

Reviewers' comments:

Reviewer's Responses to Questions

**Comments to the Author**

1. If the authors have adequately addressed your comments raised in a previous round of review and you feel that this manuscript is now acceptable for publication, you may indicate that here to bypass the “Comments to the Author” section, enter your conflict of interest statement in the “Confidential to Editor” section, and submit your "Accept" recommendation.

Reviewer #1: All comments have been addressed

Reviewer #2: All comments have been addressed

2. Is the manuscript technically sound, and do the data support the conclusions?

Reviewer #1: Yes

Reviewer #2: Yes

3. Has the statistical analysis been performed appropriately and rigorously? 

Reviewer #1: Yes

Reviewer #2: Yes

4. Have the authors made all data underlying the findings in their manuscript fully available?

Reviewer #1: Yes

Reviewer #2: No

5. Is the manuscript presented in an intelligible fashion and written in standard English?

Reviewer #1: No

Reviewer #2: Yes

6. Review Comments to the Author

Reviewer #1: (No Response)

Reviewer #2: (No Response)

7. PLOS authors have the option to publish the peer review history of their article (what does this mean?). If published, this will include your full peer review and any attached files.

Reviewer #1: No

Reviewer #2: **Yes: **Kim Madundo

---

## [Editor Report · Acceptance letter]

16 Aug 2024

PONE-D-24-07891R1 

PLOS ONE

Dear Dr. Tesema, 

I'm pleased to inform you that your manuscript has been deemed suitable for publication in PLOS ONE. Congratulations! Your manuscript is now being handed over to our production team.

Kind regards, 

on behalf of

Mr. Shegaw Tesfa Mengist 

Academic Editor

PLOS ONE